# Kinetics and mapping of Ca-driven calmodulin conformations on skeletal and cardiac muscle ryanodine receptors

Robyn T. Rebbeck [1,4] ✉, Bengt Svensson[1,4], Jingyan Zhang[1], Montserrat Samsó [2], David D. Thomas [1], Donald M. Bers [3] & Razvan L. Cornea [1] ✉

Calmodulin transduces [Ca$^{2+}$] information regulating the rhythmic Ca$^{2+}$ cycling between the sarcoplasmic reticulum and cytoplasm during contraction and relaxation in cardiac and skeletal muscle. However, the structural dynamics by which calmodulin modulates the sarcoplasmic reticulum Ca$^{2+}$ release channel, the ryanodine receptor, at physiologically relevant [Ca$^{2+}$] is unknown. Using fluorescence lifetime FRET, we resolve different structural states of calmodulin and Ca$^{2+}$-driven shifts in the conformation of calmodulin bound to ryanodine receptor. Skeletal and cardiac ryanodine receptor isoforms show different calmodulin-ryanodine receptor conformations, as well as binding and structural kinetics with 0.2-ms resolution, which reflect different functional roles of calmodulin. These FRET methods provide insight into the physiological calmodulin-ryanodine receptor structural states, revealing additional distinct structural states that complement cryo-EM models that are based on less physiological conditions. This technology will drive future studies on pathological calmodulin-ryanodine receptor interactions and dynamics with other important ryanodine receptor bound modulators.

Calcium (Ca$^{2+}$) cycling between the sarcoplasmic reticulum (SR) and the sarcoplasm is essential for muscle contraction and relaxation, which are primarily maintained by ryanodine receptors (RyR), and SR/ER Ca$^{2+}$-ATPases (SERCA pumps), respectively. Of the three mammalian RyR isoforms, RyR1 and RyR2 dominantly represent RyR in skeletal and cardiac muscle, respectively. The ~ 65% sequence identity between isoforms is consistent with the overall structural similarity of the large (2.3 MDa) homotetrameric state observed via cryo-EM. Within the last decade, cryo-EM studies have greatly enhanced our understanding of RyR structural states, resulting in near-atomic resolution structures (3.2-4.2 Å) of both RyR1 and RyR2[1–13]. Cryo-EM represents a powerful approach that has led to breakthroughs, but it comes with caveats inherent to sample preparation (non-physiological) and is limited to resolving only the most populated static structures (static or dynamic disorder is only qualitatively inferred). Also, the full RyR sequence has yet to be charted into the 3D structure.

In addition to RyR structures, cryo-EM has been used to reveal RyR co-structures with its important regulators, e.g., calmodulin (CaM)[3,4,13,14]. CaM-mediated regulation is particularly important for the proper function of RyR2, where dysregulation has been associated with cardiac hypertrophy, arrhythmia, and heart failure[15–23]. Indeed, enhancing the CaM-RyR2 interaction has been shown to restore healthy Ca$^{2+}$ cycling in cardiomyocytes[23–26]. Therefore, it is of key translational relevance to understand the mechanisms of long-range allosteric regulation of channel gating by modulators, including CaM, which bind to the large cytosolic portion of the RyRs.

[1]Department of Biochemistry, Molecular Biology & Biophysics, University of Minnesota, Minneapolis, MN, USA. [2]Department of Physiology and Biophysics, Virginia Commonwealth University, Richmond, VA, USA. [3]Department of Pharmacology, University of California at Davis, Davis, CA, USA. [4]These authors contributed equally: Robyn T. Rebbeck, Bengt Svensson. ✉e-mail: rrebbeck@umn.edu; corne002@umn.edu

Calmodulin is a 17-kDa protein that contains four EF-hands between two terminal domains (known as the N- and C-lobe). These Ca²⁺-binding domains enable finely tuned modulation of RyR during Ca²⁺ cycling. Both Apo- and Ca²⁺-bound forms of CaM have been shown to bind to RyR1 and RyR2 with a stoichiometry of four CaM per channel[14,27,28]. Ca²⁺ binding affinity ($K_{Ca}$) to CaM is enhanced from ~ 3 to ~ 1 μM by binding to RyR1 in SR membranes[29], providing Ca²⁺ sensitivity more in tune with Ca²⁺ cycling during striated muscle contraction. CaM is a well-established inhibitor of both RyR1 and RyR2 at contracting (μM) [Ca²⁺]ᵢ[30–33], thought to contribute to the termination of SR Ca²⁺ release, to allow normal cellular relaxation[19,34]. CaM also inhibits RyR2 at resting (nM) [Ca²⁺]ᵢ[32,33], which is considered protection from chronic SR Ca²⁺ leak during diastole[35]. In contrast, CaM increases RyR1 activity at resting [Ca²⁺]ᵢ[30,31], which is thought to prime the channels for activation during EC coupling[34].

Early cryo-EM structures showed that CaM binds to RyR1 in two locations depending on the Ca²⁺ concentration. At nM Ca²⁺, apo-CaM is located high in the crevice between the Handle and the Helical domain, while at μM to mM Ca²⁺, Ca-CaM bound further down, also between the Handle and Helical domains[14]. More recently, the high-resolution structures of RyR1-apo-CaM[3], RyR1-Ca-CaM[13], and RyR2 with the two forms of CaM[4] have been resolved, ratifying the distinct location of the two forms of CaM in both isoforms. In addition, the CaM binding site of RyR2 (CaM binding domain 2; CaMBD2; residues 3580−3608) was verified using peptides corresponding to this region[36,37]. Despite the distinct functional effect of CaM at resting [Ca²⁺], the high-resolution cryo-EM structures place apo-CaM on RyR2 in a similar position as for apo-CaM bound to RyR1[3]. Apo-CaM on the closed state of RyR2 binds high in the groove between the Handle and Helical domains (Fig. 1A). The N-lobe is positioned high in this groove and makes contact with both of these RyR2 domains. The C-lobe is positioned lower in the groove making contact mostly with the Handle domain, but also with the C-terminal portion of the α-helix of RyR2 (residues 3593−3606) which is part of CaMBD2[4]. Helical Domain 2 of RyR2 (residues 2982−3528) precedes the CaMBD2 region and would fill the lower portion of this groove. However, Helical Domain 2 is mostly unresolved in these RyR2 structures. Apo-CaM binds to RyR1 in the same location as previously described for RyR2, i.e., in the groove between the Handle and Helical domains with the N-lobe positioned high in this groove and making contact between these two RyR domains. The C-lobe is positioned lower in the groove, making contact mostly with the Handle Domain, but also with C-terminal portions of the CaMBD2 region of RyR1 (residues 3625−3634). Helical Domain 2 is better resolved in RyR1, but it does not make any contact with apo-CaM[3]. While the structural determinations have been carried out at 5−10 mM EGTA, and thus < 0.2 nM Ca²⁺. Without a CaM-RyR structure at physiologically [Ca²⁺] level reflective of diastolic conditions, it is difficult to determine the cause of isoform-specific regulation by apo-CaM, as apo-CaM may not represent the structural state of CaM bound to RyR in nanomolar Ca²⁺.

In the Ca²⁺-bound form, cryo-EM-based models place CaM's N-lobe shifted down on RyR2 to wrap around the canonical CaM binding site (Fig. 1B), between the Helical Domain 2 and the Central domain[4], which is consistent with a prior crystal structure of Ca-CaM bound to a RyR peptide (corresponding to RyR1 residues 3614−3643)[38]. Ca-CaM in the closed state of RyR2 is bound further down in the groove, with the two CaM lobes now wrapped around the CaMBD2 α-helix (residues 3585−3606)[4]. The N-lobe is rotated about 180° and shifted lower in the groove compared with the apo-CaM structure. The C-lobe has undergone a rotation and a shift towards Helical Domain 2. Ca-CaM on RyR1 is bound differently compared with Ca-CaM in the RyR2 structure[13]. The N-lobe is still bound high in the groove in a similar location as apo-CaM. CaM helix I is in the same area as in the apo-CaM structure but helices II and III are rotated upward toward upper Helical Domain by about 45°. CaM helix IV (which connects the two lobes) has rotated toward the RyR1 Helical Domains to allow the C-lobe a new binding location closer to Helical Domain 2. This is mostly a shift in the binding location closer to Helical Domain 2, with just a minor rotation. In contrast to RyR2, in RyR1 only the C-lobe of Ca-CaM is in contact with CaMBD2 (residues 3614−3637)[13]. As the conditions used are quite different between the RyR1 and RyR2 Ca-CaM structures, it is too speculative to conclude that the differences observed via cryo-EM in the binding of Ca-CaM are due to RyR isoform differences. Thus, for isoform comparison, it is highly desirable to resolve Ca-CaM binding in similar conditions.

We have previously used steady-state FRET to resolve the orientation of CaM lobes relative to a fluorescent probe attached to FKBP12.6 on RyR[39,40]. That type of measurement has also been used to map the divergent regions of the RyR1[41], and the RyR2 binding site of DPc10 (peptide corresponding to RyR2 residues 2460−2495)[42]. A limitation of this approach is that steady-state fluorescence measurements of FRET yield a single distance for the ensemble of observed donor-acceptor distances. To overcome this limitation, we have used fluorescence lifetime (FLT) measurements of FRET, where the nanosecond-scale decays can be analyzed to resolve up to two populations of distinct distances between the donor and acceptor[43,44]. Using this technology, we resolve here distance relationships between eight distinct probe sites on FKBP12.6 and four distinct probe sites on CaM within RyR1 and RyR2 channel complexes in SR membranes, which enriches our understanding of CaM-RyR binding in its native SR membrane in physiological solution conditions. Indeed, the resulting FRET-based structural models are developed in synergy with cryo-EM structural information to provide a physiologically relevant CaM-RyR representation. In addition to resolving structural models in an equilibrated state, the time resolution of our stopped-flow FLT-FRET system, with 2 ms mixing time and 0.2 ms temporal resolution, uniquely monitors its binding and structural kinetics on the timescale of muscle contraction. The methodology showcased here is suitable for observing the binding characteristics of many other RyR modulators.

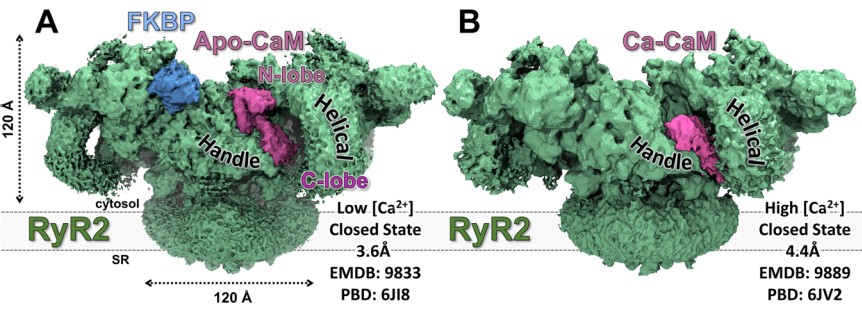

**Fig. 1 | Cryo-EM maps of RyR2 reveal CaM binding to RyR2.** Ca-free (apo) and Ca-bound -states of CaM bind at distinct locations. **A** RyR2 (green) in a closed-pore state with FKBP12.6 (Light Blue) and apo-CaM (Purple) under EGTA conditions (< 0.2 nM Ca²⁺)[4]. **B** RyR2 in a closed-pore state with Ca-CaM bound at 20 μM Ca²⁺[4].

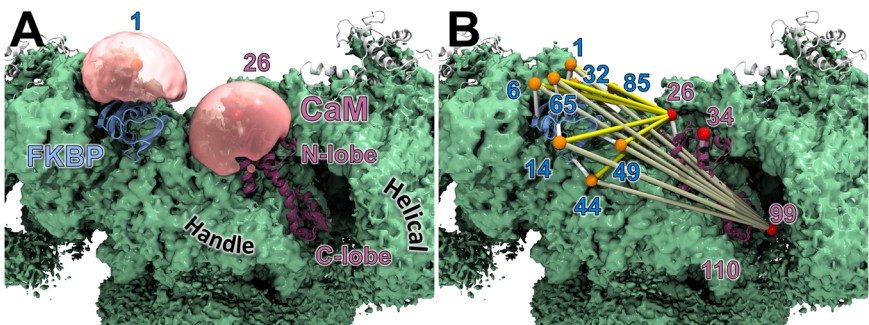

**Fig. 2 | Labeling and locations of FRET probes.** In silico predictions of the donor and acceptor fluorophore locations were conducted by simulated annealing for donors bound to FKBP, and by a method based on probability distributions for acceptors bound to CaM. Shown here are results for RyR2 (green) at low Ca²⁺ (EMDB: 9833, PDB: 6JI8)[4]. **A** The volume sampled by the fluorophore is shown as a pink bubble for the donor attached at FKBP residue 1. The average center position of the probe is shown as an orange sphere inside the pink bubble. The example CaM probe location is displayed as a pink bubble with the average acceptor location shown as a red sphere. **B** The predicted FRET distances from all donor-labeled sites on FKBP to acceptor-labeled site 26 in the N-lobe and site 99 in the C-lobe of CaM are shown with yellow and tan colored lines, respectively. The acceptor probe bound at CaM residue 110 is hidden behind the Handle domain.

## Results

### Labeling FKBP12.6 and CaM does not alter binding to RyR

For labeling CaM and FKBP12.6, we substituted single cysteine at sequence locations in CaM (without native Cys) and Cys-Null FKBP12.6[45] that would be solvent exposed and unlikely to interfere with RyR interaction (see next section in *Results*). We have previously evaluated the binding and functional effect on RyR1 of labeling FKBP12.6 with Alexa Fluor 488 C5 maleimide (AF488) at sites T14C, E32C, R49C, and T85C[39], and labeling CaM with Alexa Fluor 568 C5 maleimide (AF568) at sites T34C and T110C[45]. Here we extended our collection of single-Cys substitutions to FKBP12.6 sites G1, T6, K44, and Q65 (Supplementary Fig. 1A), and CaM sites T26 and Y99 (Supplementary Fig. 2A). Thus, we used eight AF488-labeled FKBP12.6 single-Cys mutants and four AF568-labeled CaM single-Cys mutants.

To test whether the mutation and attachment of our fluorescent FKBP12.6 probes alters their necessary high affinity binding to RyR, we performed co-sedimentation experiments of each AF488-FKBP12.6 variant with RyR1 in skeletal SR membranes and RyR2 in cardiac SR membranes. For F-FKBP12.6 binding to RyR1, the variants that were previously shown to bind similarly with WT-FKBP12.6 (T14C, E32C, R49C, and T85C)[39], we found similar binding affinities (3.7–5.8 nM $K_d$ values) for RyR1 (Supplementary Fig. 1C). Of the new single-cysteine FKBP12.6 mutants, K44C and Q65C displayed affinities within the same range, while G1C and T6C showed slightly lower affinities (12.6 and 7.3 nM, respectively) (Supplementary Fig. 1C). As expected, binding of these F-FKBP12.6 variants to RyR2 displayed similar trends (1.6–5.0 nM $K_d$ values). At 30 nM, all F-FBKP12.6 variants displayed saturated binding to both RyR1 and RyR2. Overall, these high affinities are suitable for our FRET measurements, that labeling did not disrupt the binding of FKBP12.6 to RyRs, and that such binding was saturating at 60 nM FKBP12.6 used in subsequent studies here.

To determine the functional impact of fluorescence labeling of Ca-sensitive and Ca-insensitive CaM on RyR interaction, we used [³H] ryanodine binding measurements, as previously[45]. Using the level of [³H]ryanodine binding as an index of channel activity, we show that the addition of 800 nM unlabeled WT-CaM significantly increases RyR1 activity at 30 nM Ca²⁺, and decreases RyR1 activity at 30 μM Ca²⁺ (Supplementary Fig. 2C). Addition of 800 nM unlabeled Ca-insensitive CaM (CaM₁₂₃₄) increased RyR1 activity at 30 μM Ca²⁺ and to a lesser degree at 30 nM Ca²⁺ (Supplementary Fig. 2C). These functional effects of unlabeled CaM align with previous publications[30,33,44,45]. As shown in Supplementary Fig. 2C and D, the Cys substitution and labeling of CaM does not impact the direction of functional effect at most sites. Although, labeling CaM at site T110 increased the CaM effect on RyR1 function, especially at 30 nM Ca²⁺ (Supplementary Fig. 2C). In

alignment with previous studies[44], we show that unlabeled WT-CaM decreases RyR2 activity at 30 nM and 30 μM Ca²⁺, but that effect is largely lost in CaM₁₂₃₄ (Supplementary Fig. 2D). As with RyR1, CaM labeling at sites T26, T34 and Y99 did not impact the direction of the functional effect of CaM on RyR2 (Supplementary Fig. 2). However, labeling at site T110 (as for RyR1) appears to activate RyR2 most obviously in comparison with WT-CaM at 30 nM Ca²⁺ (Supplementary Fig. 2D), although this AF568-T110C-CaM induced increase for RyR2 by ~ 38% was much less than the 20-fold increase observed even for WT-CaM on RyR1. These functional assays indicate that the fluorescence-labeled CaM mostly retained functional interactions of WT-CaM with RyR1 and RyR2 channels that are characteristic for both 30 nM and 30 μM Ca²⁺.

### Effective positions of the FKBP- and CaM-attached probes

To interpret the FKBP-CaM FRET data, we used an in silico approach to determine the location of the donor probes attached to FKBP in RyR1 and RyR2. For the computational studies of RyR1, the structure chosen was the 3.6-Å inactive state and the corresponding cryo-EM density map (Supplementary Fig. 7)[1]. For RyR2, we used the 3.6-Å closed state and the corresponding cryo-EM density map (Fig. 2, Supplementary Fig. 3)[4]. We chose the structures with closed pores because they reflect the more physiological state that predominates in our FRET samples, as discussed in *Methods*.

In the atomic structure of RyR1 and RyR2, the eight residues at positions 1, 6, 14, 32, 44, 49, 65, and 85 of FKBP12.6 that were used for probe attachment in the fluorescence experiments were mutated to cysteine in silico. The AF488 probe was attached to each of the eight cysteines, generating eight starting conformations for the simulated annealing calculations used to determine the effective donor probe coordinates. For comparison with our trilateration and to calculate predicted distances between donor and acceptor probes in the cryo-EM structure, we used a new faster method to determine the region in space the probe can sample and the average probe location on CaM at sites 26, 34, 99, and 110, as described in *Methods*.

### Resolving distance distribution between probes from FLT-FRET experiments

To map the position of CaM in 30 nM and μM Ca²⁺, we used FLT-FRET acquired using time-correlated single photon counting (TCSPC), which can be analyzed to yield distance relationships between the established donor probe sites on FKBP12.6 and acceptor probe sites on CaM (e.g., as in Fig. 3B). In each experiment, we used SR membranes from porcine skeletal or cardiac muscle that were labeled with one of the eight D-FKBP variants validated above. FLT waveforms (e.g., as in

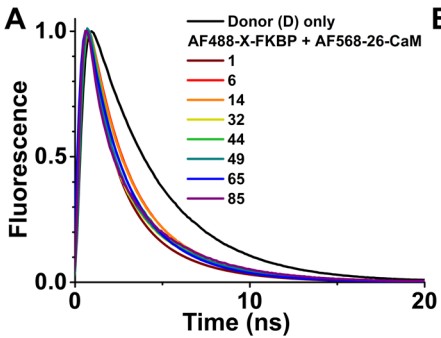
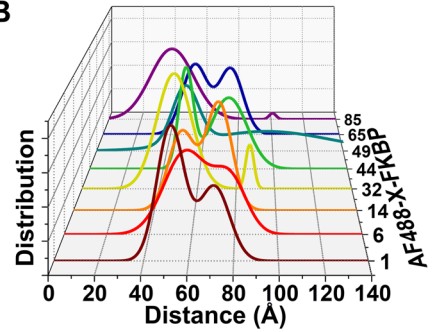

**Fig. 3 | Representative FLT-FRET detection of distance distributions between D-FKBP and A-CaM probes.** At 30 nM $Ca^{2+}$, SR membranes from skeletal muscle were labeled with D-FKBP (AF488-X-FKBP; $X$ = Cys mutation site for fluorescence labeling), and then incubated with 800 nM A-CaM (labeled with acceptor probe at the N-lobe residue T26C). For each label site on FKBP, site 1 is maroon, site 6 is red, site 14 is orange, site 32 is yellow, site 44 is green, site 49 is teal, site 65 is blue and site 85 is purple. **A** Representative FLT-detected waveforms for each of the eight

different D-FKBP. These waveforms were acquired at least three times with similar results. **B** Multi-exponential analysis of FLT-FRET data yielded a two-distance Gaussian distribution model for the separation between D-FKBP and A-CaM within RyR1. Averaged data shown in Supplementary Fig. 3. These waveforms were acquired at least three times with similar results. Source data are provided as a Source Data file.

Fig. 3A) were acquired 2 h after incubation of our FRET samples with A-CaM or A-$CaM_{1234}$, both with 4 variations by being labeled at one of four sites, as described above. We have previously used single distances calculated from FRET efficiencies to locate acceptor-labeled sites on RyR cryo-EM maps[41,42]. Here, we enhance this method, as using fluorescence decays allows for simultaneous detection of CaM and FKBP12.6 binding to RyR and structural analysis. This approach allows us to integrate the fraction of unbound donors (~10%; not participating in FRET) in the distance calculations and thus eliminate their contribution[44,45]. Furthermore, with the possibility of FRET from neighboring faces of the tetrameric RyR complex, extracting multiple-distance information enhances the resolution of the actual distance relationships between D-FKBP and A-CaM bound on the same face of RyR.

With the CaM N-lobe probe locations being closer to probes on FKBP12.6 (Fig. 1), a two-distance Gaussian distribution model of separation between D-FKBP and A-CaM represented the best fit of the FLT-FRET data (Fig. 3). When the second distance is comparable to the donor-acceptor $R_0$ (60–80 Å), it is probably due to the co-existence of at least two significantly populated conformational states of the D-FKBP/A-CaM/RyR complex. Our results indicate a high probability that intermediate CaM binding states exist. When the second distance is long (around 100 Å, e.g. top curve in Fig. 3B), it may be due to FRET with A-CaM on a neighboring protomer. Distance predictions based on cryo-EM derived models show that, for the N-lobe CaM analysis, the next closest CaM on an adjacent RyR protomer is >115 Å from FKBP12.6, on the next closest protomer >160 Å, and on the furthest protomer >195 Å away. Such distances, beyond 120 Å, are beyond the detection range for our donor-acceptor probes, and therefore A-CaMs in those positions cannot contribute to the FRET measurement. Considering this, our distance analysis focuses on the most significant component. In contrast, FRET between D-FKBP and A-CaM with the probe attached at the C-lobe produced only a longer, single-distance Gaussian distribution. For some of the FRET pairs, cryo-EM-based models suggest that the distance to CaM on the adjacent protomer is just slightly longer than to the closest CaM on the same protomer and can also contribute to the observed FRET. This and the longer distance to the C-lobe sites may explain why we cannot resolve more than one distance distribution in our fitting of these FRET data. Two acceptors that both contribute to FRET may lead to a slight underestimation of the FRET distances. These results align with our previous report using the same model with AF488-R49C-FKBP12.6 and N-lobe labeled CaM (T34C) vs C-lobe labeled CaM (T110C)[44].

The averaged parameters of each Gaussian fit are shown in Supplementary Figs. 3–6. This type of data was used for trilateration, as shown in Supplementary Tables 1–4.

## Trilateration of CaM binding on RyR 3D map demonstrates $Ca^{2+}$-driven structural shifts

The method of using distances to determine a location in space is termed trilateration. This method is most commonly known for its use in the Global Positioning System (GPS), where known satellite-positions are used to determine an antenna location. The use of trilateration based on FRET distances and macromolecules has been previously described by us and others[42,46,47]. From the simulated-annealing calculations above, we obtained coordinates of the effective donor locations, which we used for trilateration. We used effective AF488 locations calculated for each of the eight FKBP attachment sites (1, 6, 14, 32, 44, 49, 65, and 85) together with Gaussian distance distributions calculated from FLT-FRET analysis, to determine a locus in space corresponding to acceptors attached at each of the four locations on CaM (26, 34, 99, and 110). In most cases, the shortest distance was the most populated, but there were a few cases where the longer distance was more populated. Upon comparison of acceptor loci sites, we chose to trilaterate using the shortest distance, except when the longer-distance population was more than double the shorter-distance population (Supplementary Fig. 8).

As expected, all acceptors were localized to the grove between the Handle and Helical domain on RyR, with the CaM N-lobe oriented above its C-lobe (Figs. 1 and 4). To differentiate between $Ca^{2+}$-driven structural shifts in RyR vs CaM, we compared the loci of probes bound to $CaM_{1234}$ in nM and μM $Ca^{2+}$. The acceptor loci for $CaM_{1234}$ did not shift between nM and μM $Ca^{2+}$, suggesting that any $Ca^{2+}$-driven shift observed with probes bound to normal CaM is due to its own structural shift, and not of RyR.

The probe locus of AF568-26C-CaM on RyR1 was driven down and outwards by the nM to μM shift in $Ca^{2+}$ (Fig. 4A and Supplementary Fig. 9). The probe locus of AF568-34C-CaM (also in the N-lobe) did not detectably shift between nM and μM [$Ca^{2+}$] for both CaM and $CaM_{1234}$ (Fig. 4A and Supplementary Fig. 9). Together, these suggest that $Ca^{2+}$ binding to CaM structurally results in a rotation of the N-lobe as it shifts downwards and out within the RyR1 binding groove. Similar to the probe at residue 34, the probe at CaM residue 99 showed a small shift for both CaM and $CaM_{1234}$ (Fig. 4A and Supplementary Fig. 9). In contrast, the probe locus of AF568-110C-CaM was distinctly shifted upward and inward in μM $Ca^{2+}$ relative to nM $Ca^{2+}$. For both residue 26 and 110 acceptor attachment sites in CaM, the loci in nM $Ca^{2+}$ overlap

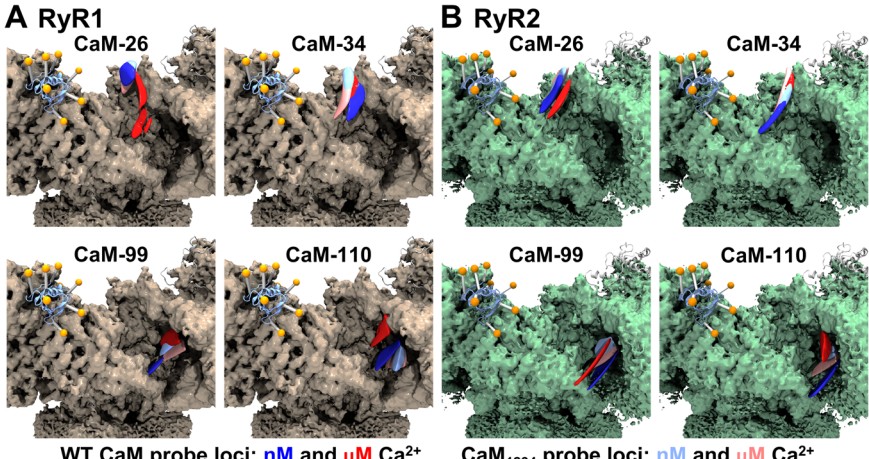

**Fig. 4 | Location of trilaterated acceptor fluorophore loci on RyR1 and RyR2.** Cryo-EM maps of (**A**) RyR1 (EMDB: 8342, PDB: 5T15)[1] and (**B**) RyR2 (EMDB: 9833, PDB: 6JI8)[4] bound to the atomic structure of FKBP (orange), with trilaterated loci of AF568 attached to CaM residues as indicated. Trilaterated loci of probes bound to CaM in assay conditions containing 30 nM and 30 μM free Ca²⁺ are represented in blue and red, respectively. Trilaterated loci for probes bound to Ca²⁺ insensitive CaM (CaM₁₂₃₄) in assay conditions containing 30 nM and 30 μM free Ca²⁺ are represented in light blue and light red, respectively.

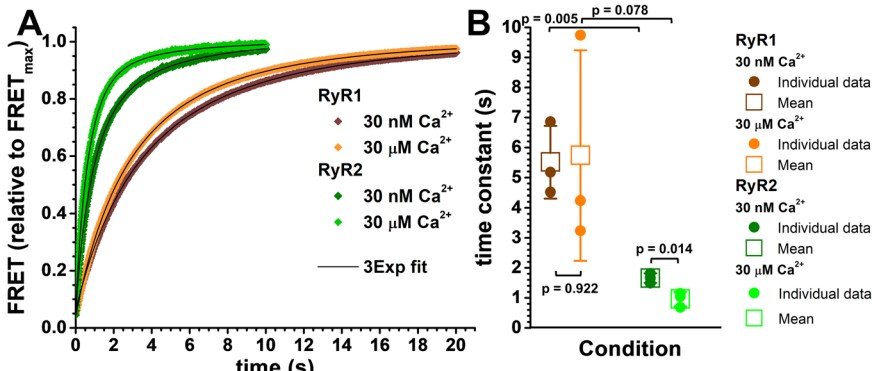

**Fig. 5 | Kinetics of A-CaM binding to RyR1 and RyR2 in 30 nM and μM Ca²⁺.** SR membranes from porcine skeletal (brown) or cardiac (green) muscle were labeled with D-FKBP (AF488-85-FKBP), and then FLT time course was acquired after rapid (2 ms) mixing with 800 nM (final) A-CaM (AF568-26-CaM), in 30 nM Ca²⁺(dark brown for RyR1 and dark green for RyR2) or 30 μM Ca²⁺ (light brown for RyR1 and light green for RyR2). **A** Representative FRET time course following mixing. The FRET time courses were acquired in three independent experiments with similar results. **B** Amplitude-weighted average time constant from three-exponential analysis of FLT-FRET data for CaM binding to RyR1 or RyR2 at 30 nM or 30 μM Ca²⁺. Data is shown as mean (open square) ±SD, $n = 3$ individual experiments (filled circle) prepared on different days. Significance calculated using unpaired, two-way Student's $T$ test. Parameters from three-component fitting are shown in Supplementary Fig. 13. Source data are provided as a Source Data file.

with the corresponding loci of CaM₁₂₃₄ at both nM and μM Ca²⁺ (Fig. 4A and Supplementary Fig. 9). This suggests that at nM Ca²⁺, both CaM lobes reflect an apo (Ca²⁺ unbound) structural state. Overall, the nM to μM Ca²⁺ shift appears to drive RyR1-bound CaM to form a more compact state, with the lobes rotating in opposite directions and driving closer together.

The probe locus of AF568-26C-CaM on RyR2 displays a similar Ca²⁺ driven downward and outward shift as we observed with RyR1, although more subtle. At nM Ca²⁺, this locus in normal CaM is positioned similarly to CaM₁₂₃₄ (Fig. 4B and Supplementary Fig. 9). In contrast to RyR1-bound CaM, the probe locus of AF568-34C-CaM in RyR2 is similar between normal CaM and CaM₁₂₃₄, but distinctly shifted lower in nM relative to μM Ca²⁺ (Fig. 4B and Supplementary Fig. 9). Acceptor loci at both CaM-99 and CaM-110 display an upward shift due to nM to μM [Ca²⁺]. Additionally, the CaM-110 probe locus appears to shift inward between nM and μM Ca²⁺ (Fig. 4B and Supplementary Fig. 9). Similar to RyR1-bound CaM, the nM to μM Ca²⁺ shift appears to drive RyR2-bound CaM to form a more compact state, with the lobes rotating in opposite directions and moving closer together.

## Kinetics of CaM-RyR binding using stopped-flow FLT-FRET

An advantage of direct waveform recorded FLT-FRET technology is its compatibility with stopped-flow kinetic studies, as we previously reported with myosin-focused, transient structure studies[48,49]. Armed with the probe pair that displayed the largest distance difference between nM and μM Ca, D-85-FKBP, and A-26-CaM (Supplementary Table 1 and 3), we carried out stopped-flow FLT-FRET measurements to first investigate CaM binding. We intended to explore the capabilities of this assay and technology. With 2-ms mixing time and waveforms acquired every 0.2 ms, this technology enabled the measurements of CaM-RyR association kinetics, and a direct comparison between the skeletal and cardiac RyR isoforms from porcine tissue. This complements current knowledge in the field of full-length RyR isoform (RyR1 vs RyR2) impact on CaM dissociation rates[28,50].

As above with our trilateration studies, to resolve differences between apo- and Ca-CaM, we concurrently monitored the binding of CaM₁₂₃₄ to RyR1 and RyR2 in 30 nM and μM Ca²⁺. In all conditions, the time courses are best fit using three exponentials (Fig. 5 and Supplementary Figs. 12 and 13). As shown in Fig. 5, the amplitude-weighted

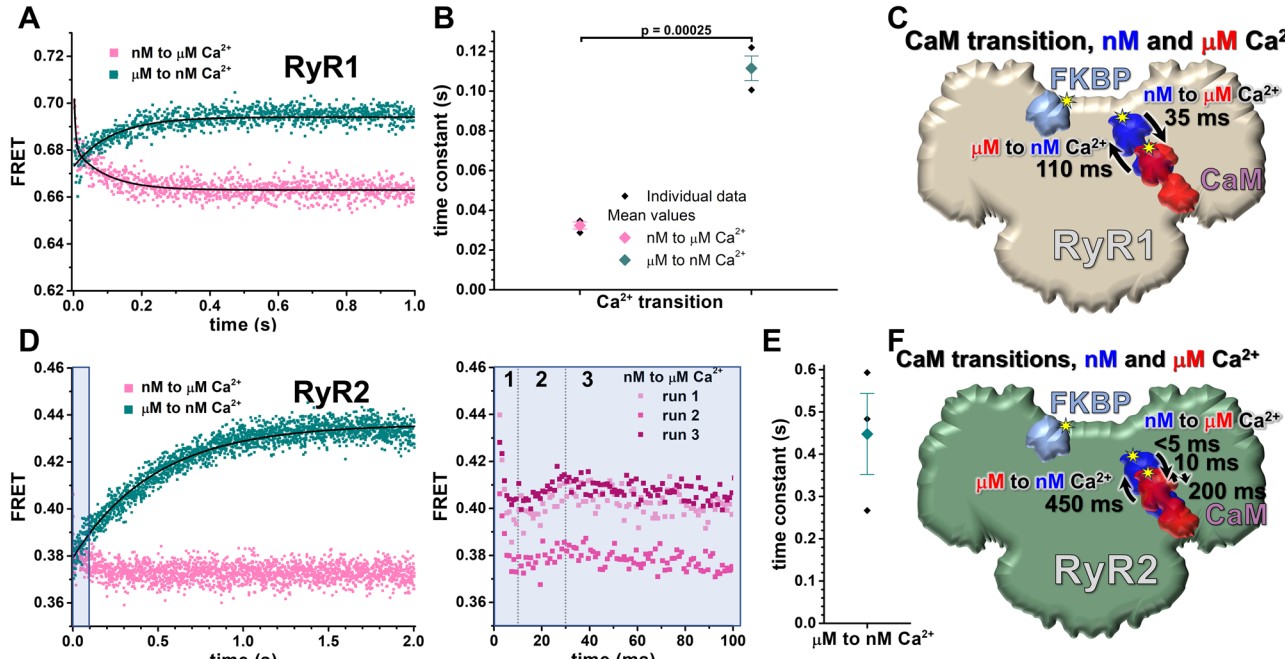

**Fig. 6 | Calcium-driven structural transitions of A-CaM bound to RyR1 or RyR2.** SR membranes from porcine skeletal and cardiac muscle were labeled with D-FKBP (AF488-85-FKBP), incubated with 1.6 μM A-CaM, and then FLT time course was acquired after rapid (2 ms) mixing with $Ca^{2+}$ to increase $[Ca^{2+}]$ from 30 nM to μM (pink), or EGTA to decrease $[Ca^{2+}]$ from 30 μM to nM (green). Representative FLT waveforms are shown in Supplementary Fig. 14. **A** With A-CaM bound to RyR1, representative FLT-FRET time course following rapid $[Ca^{2+}]$ shift. FRET data for nM to μM $Ca^{2+}$ transition fit best to two-exponential analysis with parameters shown in Supplementary Fig. 15. For μM to nM, $Ca^{2+}$ transition fit best to one-exponential function. The FRET time courses were acquired in three independent experiments with similar results. **B** Averaged time constants per $Ca^{2+}$ transition. Data shown as mean (closed diamond) ± SD, $n = 3$ individual (solid black diamond) experiments prepared on different days. Significance calculated using unpaired, two-way Student's $T$ test. **C** Schematic of structural transition of CaM (30 nM $Ca^{2+}$ in blue, 30 μM $Ca^{2+}$ in red) on RyR1 (light brown) relative to FKBP (light blue) based on PDBs 6JI8 and 6JV2, with time constants based on $Ca^{2+}$-driven FRET shift. **D** With A-CaM bound to RyR2, representative FRET time courses following rapid $[Ca^{2+}]$ shift. For μM to nM, $Ca^{2+}$ transition fits best to one-exponential fitting. The FRET time courses were acquired in three independent experiments with similar results. Within the first tenth of a second, FRET data for nM to μM $Ca^{2+}$ transition was multiphasic, which was shown to be reproducible, as illustrated in the right panel, where each phase 10 ms, 10–30 ms and > 30 ms is indicated by numbers 1, 2 and 3, respectively. **E** Time constant for μM to nM $Ca^{2+}$ transition for A-CaM bound to RyR2. Data are shown as mean (closed diamond) ± SD, $n = 3$ individual experiments (solid black diamond) prepared on different days. **F** Schematic of structural transition of CaM (30 nM $Ca^{2+}$ in blue, 30 μM $Ca^{2+}$ in red) on RyR2 (green) relative to FKBP (light blue) based on PDBs 6JI8 and 6JV2, with time constants based on $Ca^{2+}$-driven FRET shift. Source data are provided as a Source Data file.

time constants for RyR2 are 3.3- and 6-fold faster than RyR1 at 30 nM and μM $Ca^{2+}$, respectively. Given that the CaM binding to either RyR1 or RyR2 measured in Fig. 5 is very much faster than the dissociation constant (by > 30-fold)[21,28,51], these time constants for CaM binding (at 800 nM CaM) are closely indicative of CaM-RyR association kinetics (e.g. an apparent $k_{on}$ of $75 \times 10^6 M^{-1} min^{-1}$ for RyR2 at nM $Ca^{2+}$). Our results indicate that CaM-RyR1 association kinetics does not significantly differ between relaxing (nM $Ca^{2+}$) and contracting (μM $Ca^{2+}$) conditions (Fig. 5). The $CaM_{1234}$-RyR1 association kinetics under both $Ca^{2+}$ conditions was similar to that of CaM-RyR1 (Supplementary Fig. 13). In contrast, the CaM-RyR2 association was significantly (~ 2-fold) faster at 30 μM $Ca^{2+}$ relative to CaM at 30 nM $Ca^{2+}$ and $CaM_{1234}$ at both $Ca^{2+}$ (Supplementary Fig. 13). Overall, this indicates that CaM binds faster to RyR2 vs RyR1, and that binding is $[Ca^{2+}]$ sensitive for RyR2 but not for RyR1.

### Kinetics of Ca-driven structural shift of RyR-bound CaM

With distinct $Ca^{2+}$-dependent distances between D-85-FKBP and A-26-CaM (Supplementary Table 1 and 3), we proceeded to use this probe pair to observe the kinetics of Ca-driven A-CaM structural transitions. Specifically, focusing on the 30 nM to μM and 30 μM to nM $Ca^{2+}$ transitions, and with saturating CaM concentrations (0.8 μM) to assure changes were mediated by altered CaM position, rather than changes in the level of CaM saturation of RyR.

For A-CaM bound to RyR1, we show that the structural transition is faster (3.4-fold) going from nM to μM $Ca^{2+}$ (35 ms, amplitude-weighted average from 4 and 79 ms in Supplementary Fig. 15), relative to the μM to nM $Ca^{2+}$ transition (110 ms; Fig. 6A, B). For A-CaM bound to RyR2, the μM to nM Ca-driven structural transition, is 4-fold slower ($p = 0.025$) relative to A-CaM bound to RyR1 (450 vs. 110 ms; Fig. 6C, E).

With the 30 nM to μM $Ca^{2+}$ transition, A-CaM bound to RyR2 undergoes a multiphase shift in FRET (Fig. 6D, right panel). FRET initially drops very rapidly within the first 10 ms (stage 1), followed by a FRET increase from 10 to 30 ms, and ending with a slower FRET decrease beyond 30 ms (Fig. 6D, right panel). With saturating A-CaM bound and slower association kinetics, these $Ca^{2+}$-dependent FRET changes might reflect stepwise structural changes of CaM as it changes conformation and shifts its position relative to RyR2 (Fig. 6F).

RyR isoform comparison for the nM to μM $Ca^{2+}$ driven shift in bound CaM shows that stage 1 and stage 3 of CaM-RyR2 kinetics is somewhat reflected in the CaM-RyR1 kinetics (4 and 79 ms time constants, Supplementary Fig. 15). With FRET decreasing in both stages, this indicates that Ca-association is shifting the N-lobe of CaM down further in the cleft between Helical and Handle domains of RyRs (Fig. 6F). However, with only RyR2-bound CaM, we observe an additional intermediate (2nd) stage with FRET increasing, indicating that N-lobe of RyR2-bound CaM briefly shifts upwards in the cleft between Helical and Handle domains of RyR2 during the Ca-driven structural transition.

## Discussion

We have advanced our FRET-based technology for molecular trilateration to map Ca-driven structural transitions of RyR-bound CaM in

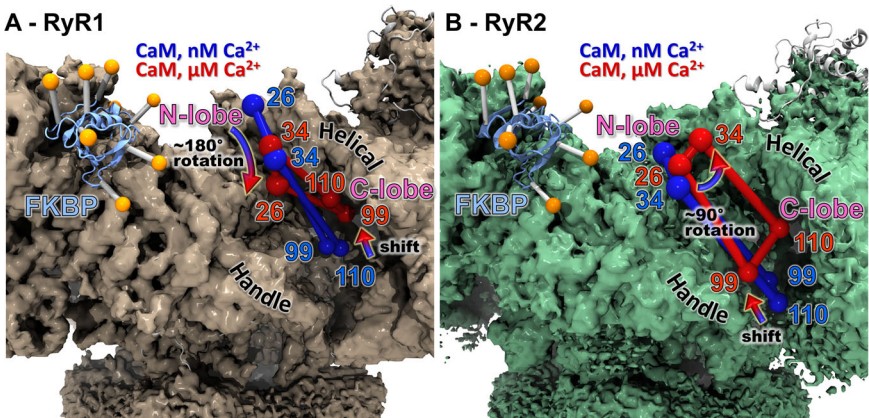

**Fig. 7 | Overall Ca²⁺ driven structural transitions of A-CaM probe loci on RyR1 or RyR2.** The center of the trilaterated loci for AF568 probes bound to indicated CaM residues are shown as spheres, drawn in blue for nM Ca²⁺ (apo-CaM) and red for µM Ca²⁺ (Ca-CaM). The spheres are connected by lines to better show the shape of and change in rotation of sites in the lobes of CaM. **A** The cryo-EM density map of RyR1[1] is shown in brown. FKBP is shown as a ribbon representation in blue with the donor probe positions shown as orange spheres. A Ca²⁺ induced 180° rotation and a shift downward is observed in the N-lobe and a shift upward is observed in the C-lobe. **B** The cryo-EM density map of RyR2[4] is shown in green. FKBP is shown as a ribbon representation in light blue with the donor probe positions shown as orange spheres. A Ca²⁺ induced ~90° rotation is observed in the N-lobe and a minor shift upward is observed in the C-lobe. Arrows indicate the direction and rotation of structural shift between nM and µM Ca²⁺.

native SR membranes, under physiologically relevant conditions. Furthermore, we have exploited rapid FLT data acquisition – as enabled by direct-wave recording technology implemented in a stopped-flow apparatus – for breakthrough observations of this structural transition's kinetics at millisecond time-resolution. Existing cryo-EM maps provide important constraints for our analyses. Although extremely informative for the overall understanding of the RyRs, the cryo-EM reconstructions come with the significant caveat that they describe static snapshots of the most populated structural states of the RyR channel complex, after removal from its physiological environment. Our initial stopped-flow kinetics observations provide unique insight into the association of CaM to RyR1 vs RyR2 – for physiologically relevant states and time scales – thus completing the description of CaM-RyR binding kinetics that had started with previous studies of the much slower CaM dissociation from RyR1 vs RyR2[50].

By combining the trilaterated locations of the A-CaM probes (Fig. 4) we can determine the binding locations and obtain information on the orientation of the individual lobes of CaM at low and high [Ca²⁺] in both RyR1 and RyR2. This allows us to directly compare the CaM binding position in our studies vs. the published cryo-EM structures. At µM Ca²⁺, FRET trilateration places the N-lobe of apo-CaM high in the groove between the Handle and Helical domains, with position 26 higher up than 34 in both RyR1 and RyR2 (Fig. 7 and Supplementary Fig. 16). The C-lobe sites are observed lower in this groove, close to the Handle Domain. These locations overlap with those shown in the cryo-EM structures for both RyR1 and RyR2, comparable with the apo-CaM position of RyR2 in Fig. 1.

In RyR1 at µM Ca²⁺ relative to nM Ca²⁺, we observe a large downward shift of position 26 while position 34 shows a much smaller shift. This suggests a Ca²⁺-driven rotation of the N-lobe by about 180°, which is distinct from the ~90° observed in current cryo-EM models[3,13] (Supplementary Fig. 16). With FRET, we also see a significant upward shift of the C-lobe sites towards the Helical domains. Combined, these suggest that Ca-CaM is in a more compact conformation than apo-CaM. However, this is quite different from what is seen in the RyR1/Ca-CaM cryo-EM structure (7TZC)[13]. Rather, our results with RyR1 are similar to what has been reported for RyR2/Ca-CaM cryo-EM structure (6JV2)[4]. This discrepancy between FRET-based trilateration and cryo-EM suggests the existence of Ca-CaM/RyR1 structural states that are invisible to cryo-EM, potentially due to disorder or sample preparation conditions.

In RyR2 at µM Ca²⁺ relative to nM Ca²⁺, we observe a minor downward shift of position 26 and a shift towards the Helical domains for position 34. This suggests a Ca²⁺-driven rotation of the N-lobe by about 90°, which is smaller than the ~180° observed in current cryo-EM models (Supplementary Fig. 16). With FRET, we see an upward shift of the C-lobe sites, albeit much smaller in RyR2 than what was observed in RyR1. Position 110 shifts closer to the Helical domains. Ca-CaM is more compact than apo-CaM but not as compact as seen in RyR1. Therefore, the FRET trilateration suggests sharply different structural shifts relative to the RyR2/Ca-CaM cryo-EM structure (6JV2)[4], which has a very compact Ca-CaM bound lower in the groove (Supplementary Fig. 16). The FRET-trilaterated Ca-CaM RyR2 binding location is similar to what is seen for Ca-CaM in the RyR1 cryo-EM structure (7TZC)[13]. Conversely, our RyR1 trilateration results suggest a more compact CaM in response to Ca²⁺, which is not observed in the RyR1 cryo-EM structure. Our results suggest a Ca-CaM RyR2 conformation that is between the conformations seen in the RyR1 and RyR2 cryo-EM structures. A possible interpretation of the trilateration results is that, under the more physiological conditions allowed by the FRET measurement, multiple locations of Ca-CaM coexist in the RyR2 CaM-binding site. It is known that mobile regions of a protein are challenging for proper modeling in cryo-EM due to their poor resolution. Thus, it is possible that for multiple conformations, FRET captured a more representative average coming from all conformations, whereas cryo-EM represents the conformation leading to the highest resolution. The combination of both techniques is thus highly valuable as it provides high-resolution data while reflecting the dynamic nature of the interactions between RyR and CaM in physiological conditions.

The FRET-based trilateration results depict RyR isoform-specific structural transitions of CaM that are consistent with its divergent regulation of RyR1 (activator in nM Ca²⁺, inhibitor in µM Ca²⁺), vs. inhibition of RyR2 at all physiologic Ca²⁺.

In conclusion, our FRET trilateration can accurately locate the binding and orientation of CaM in RyR. Overall, our results match the position and orientation of CaM in the cryo-EM structures, especially for apo-CaM. Furthermore, our results suggest that there are different binding locations for Ca-CaM than reported in the current cryo-EM structures. One could speculate that if the sample conditions were unified between cryo-EM and FRET, and cryo-EM datasets were sufficiently large to allow 3D reconstructions of all existing conformations, similar Ca-CaM binding would be observed. At this point, insufficient

cryo-EM data exists to unambiguously describe the RyR isoform-specific differences in Ca-CaM binding. Our stopped-flow FRET kinetics results further support the existence of these differences.

Before this study, knowledge of the temporal relationships between CaM and RyR was limited, and mostly involved peptides corresponding to CaM binding domains of RyR1/2[36,37,50,52]. Here, we used a stopped-flow apparatus with our FRET assay, to monitor the fast association rates of CaM binding to RyR1/2, as well as the faster rates of $Ca^{2+}$-driven structural transition in CaM bound to RyR1/2. We provide evidence suggesting that the observed FRET changes are due to direct shifts in CaM binding to RyR through the CaM N-lobe, where the acceptor probe is bound. Due to low FRET and smaller Ca-driven structural shifts, we were unable to resolve the kinetics of A-CaM binding and Ca-driven structural transition through monitoring FRET with the acceptor probe bound at the CaM C-lobe. This reduced resolution is probably due to the method of FLT data acquisition, as direct waveform recording was used for acquiring fluorescence decays in the kinetics experiments, whereas the more resolved but much slower FLT acquisition technology (time-correlated single photon counting) was used for the acquisition of fluorescence decays used in trilateration (at equilibrium). An additional limitation of the kinetics measurements is that we could not confidently fit the Gaussian distance distributions to the kinetics data, as we did with the data used for trilateration.

Our results show that the CaM-RyR1 association kinetics does not change between nM and µM $Ca^{2+}$. This was surprising given that the functional effects of CaM on RyR1 differ so sharply between the two $[Ca^{2+}]$ tested. In contrast, we show that CaM-RyR2 association kinetics is 2-fold faster for µM relative to nM $Ca^{2+}$, which is consistent with the greater CaM-mediated inhibition of RyR2 at µM relative to nM $Ca^{2+}$. In comparison between RyR1/2 isoforms, we also observed a substantially faster CaM association with RyR2 relative to its association with RyR1. This aligns with a previous report that CaM dissociation from RyR2 is 3-fold faster than from RyR1[50]. In contrast to our observations with full-length RyR in native SR membrane preparations, a recent study using peptides corresponding closely to the highly conserved CaMBD2 of RyR1 (residues 3614-3640) and RyR2 (residues 3580–3608) showed that the rates of CaM association with RyR1 and RyR2 peptides did not significantly differ in 5 mM $Ca^{2+}$ [37]. This suggests that the RyR isoform difference that we observe is due to regions outside CaMBD2, including accessibility to the CaMBD2 site in the context of the full-length protein, and further highlights the importance of using full-length RyR in CaM-RyR studies.

CaM association kinetics do not differ between normal CaM and $CaM_{1234}$ for binding to RyR2 at the same nM $Ca^{2+}$. This supports our trilateration results in that CaM is likely in an apo-state when binding to RyR2 in 30 nM $Ca^{2+}$ and allows time for equilibration.

By monitoring CaM structural transitions in response to rapid changes between nanomolar and micromolar $[Ca^{2+}]$, we determined temporal correlations with CaM-RyR functional modulation in skeletal and cardiac muscle during EC coupling and relaxation. With both 30 nM to µM and 30 µM to nM $Ca^{2+}$, we observed A-CaM shifts on RyR1 substantially (>50-fold) faster than the observed for the A-CaM binding to RyR1 in nM and µM $Ca^{2+}$. This suggests that the FRET shift over time can be attributed to a structural transition while CaM remains bound to RyR1, rather than to changes caused by A-CaM binding to RyR1. Note that the FRET shift was 3.4 ± 0.1 times faster for the $[Ca^{2+}]$ shift from nM to µM relative to that from µM to nM (Fig. 6). Extrapolating to physiology, this suggests that CaM shifts rapidly during EC coupling to becoming an RyR1 inhibitor, faster than its transition to becoming an RyR1 activator during relaxation.

In contrast, the $Ca^{2+}$ driven structural shift in A-CaM bound to RyR2 appears to be more complex. The A-CaM with RyR2 FRET changes driven by rapid $[Ca^{2+}]$ changes are >10 times faster than the CaM-RyR2 association kinetics and even faster when compared to CaM

dissociation kinetics. Thus, these shifts are consistent with changes in CaM's structural state on RyR2, rather than in overall CaM-RyR2 association. For A-CaM bound to RyR2, the micromolar to nanomolar $Ca^{2+}$-driven structural transition, is 4 times slower relative to A-CaM bound to RyR1, indicative of a higher affinity Ca-bound state of CaM on RyR2.

The kinetics of CaM movement on RyR2 is observed as the slow upward movement of CaM upon the rapid decline of $[Ca^{2+}]$, with a time constant of 450 ms. This is on the time scale of the diastolic interval in the heart. This movement of CaM on RyR2 could thus parallel (with some delay) the decline in intracellular $[Ca^{2+}]$ and relaxation during the heartbeat and might reflect partial $Ca^{2+}$ dissociation from the RyR2-bound CaM (possibly in the N-terminal $Ca^{2+}$ binding sites) and resultant repositioning on RyR2. Thus, CaM may not dissociate from the RyR2 between beats but may shift location on the RyR2. When local $[Ca^{2+}]$ around the RyR2 rises rapidly, as during SR Ca release, the CaM movement is fast (nearly complete within 100 ms), but is multiphasic, suggesting intermediate structural steps that may include rapid $Ca^{2+}$ binding to CaM and also CaM repositioning on RyR2. The first stage is very fast, like the peak of SR Ca release (within 10 ms), and a large fraction of the overall movement. Stage 2 during the 10−30 ms time frame is in the opposite direction and is unique to RyR2 (not seen in RyR1). The functional role of stage 2 is unclear, but raises some new questions, which could be revealed by further testing with conditions that shift CaM-RyR2 binding, such as pathologic states of RyR2/CaM.

Overall, we presented the possible locations of the individual lobes of CaM bound to RyR under nM and µM $Ca^{2+}$ conditions based on trilateration using FRET distances. Our data indicates that the cryo-EM models of apo-CaM bound to RyR1 and 2 are likely representative of the physiological states. In contrast, our trilaterated Ca-CaM-RyR2 models yield structures that are distinct from the cryo-EM maps, thus indicating that further cryo-EM studies will be required to unambiguously describe the RyR isoform-specific differences in Ca-CaM binding under physiological conditions. Alternatively, MD simulations guided by our FRET distances (Supplementary tables 1–4) can be used to gain insight on intermediate locations for CaM binding to RyR. Indeed, our kinetics studies highlight intermediate states of $Ca^{2+}$-driven structural transitions of RyR-bound CaM, with a distinct state for RyR2-bound CaM. This technology sets a path for further testing of pathological implications of structural RyR regulation by CaM and other modulators.

## Methods

The authors confirm that their research complies with all relevant ethical regulations. This study does not include research with live animals or humans.

### Expression, purification, and fluorescent labeling of FKBP and CaM

Human FKBP12.6 cDNA was modified by site-directed mutagenesis (QuikChange Lightning kit cat 200518; Agilent Technologies) to introduce one of eight single cysteine mutants (G1C, T14C, E32C, K44C, R49C, Q65C, T85C) into a null cysteine background (C22A and C76I FKBP12.6). Mutant FKBPs were expressed in *Escherichia coli* BL21(DE3) pLysS, purified, and labeled with AlexaFluor488 (ThermoFisher Scientific, cat# A10254) as previously undertaken[39]. A pRSETb T7 expression vector plasmid inserted with human FKBP12.6 variant cDNA was transformed into *E. coli* BL21(DE3)pLysS (Promega, Wisconsin, USA) and the cells were grown at 37 °C until the optical density at 600 nm reached ~1. At that point, protein expression was induced by the addition of 0.4 mM isopropyl β-D-thiogalactoside and further 12−16 h incubation at 21 °C. The cells were spun, resuspended in a homogenization solution (2 mM EDTA, 1 mM DTT, 1 mM PMSF, 1 mM benzamidine, 0.1% TWEEN 20, 31 µg/mL DNaseI, 5 mM $MgCl_2$ and 50 mM Tris-HCl, pH 7.5) and briefly (2.5 min total time) sonicated on

ice with a Branson Ultrasonics Sonifier 250 (Danbury, CT, USA). The homogenate was spun at 10,000 x $g$ for 30 min. The supernatant was incubated for 20 min at 4 °C with 45% saturated ammonium sulfate. Following centrifugation, the ammonium sulfate saturation was increased to 60% and the supernatant was incubated for 20 min at 4 °C and then centrifuged. The pellet was resuspended in a *size exclusion chromatography* buffer containing 10 mM HEPES, 150 mM NaCl, 0.2 mM TCEP, 1 mM EDTA, 0.02% NaN$_3$, 10 mM HEPES (pH 8.0). The FKBP12.6 variant was further purified by loading onto a HiPrep 26/60 Sephacryl S-100 HR column (MilliporeSigma, Darmstadt Germany) and eluted using the *size exclusion chromatography* buffer. SDS-PAGE and Coomassie staining were used to identify FKBP12.6-containing fractions. These fractions were pooled, dialyzed in 30 mM NaCl and 20 mM MOPS (pH 7.0) and concentrated using an Amicon Stirred Cell with a 3 kDa Ultrafiltration disc (EMD Millipore, Billerica MA). For dye labeling, degassed 0.087 mM FKBP12.6 protein variant was incubated for 3 h at 22 °C with 0.694 mM AlexaFluor 488 C5 maleimide in solution containing 5 mM EDTA, 1 mM TCEP, 10% dimethylformamide, 100 mM TRIS·HCl (pH 7.5). Before binding to a column containing Phenyl Sepharose CL-4B (GE Healthcare Bio-Sciences AB, Uppsala Sweden), the labeling reaction was diluted 10-fold in a solution containing 35% saturated ammonium sulfate, 3% dimethylformamide, and 30 mM TRIS·HCl (pH 7.5). Following washing away the free dye with a dilution solution, the labeled protein was eluted by loading a solution containing 50 mM NaCl, 5% dimethylformamide, and 20 mM TRIS·HCl (pH 7.5). Fractions containing labeled protein were dialyzed in 30 mM NaCl and 20 mM MOPS (pH 7.0) and stored at −80 °C.

Single-cysteine CaM with substitutions within either N lobe (T26C, T34C) or C lobe (Y99C and T110C) were expressed, purified and labeled with AF568 C5 maleimide as previously[29]. A pET-7 T7 expression vector plasmid inserted with human CaM variant cDNA was transformed into *E. coli* BL21(DE3)pLysS, and the cells were grown at 37 °C until optical density at 600 nm reached ~ 1. At that point, protein expression was induced by the addition of 0.4 mM isopropyl β-$_D$-thiogalactoside and a further 4 h incubation. The cells were spun, resuspended in a homogenization solution (2 mM EDTA, 1 mM DTT, 1 mM PMSF, and 50 mM Tris-HCl, pH 7.5) and briefly (2.5 min total time) sonicated on ice with a Branson Ultrasonics Sonifier 250 (Danbury, CT, USA). The homogenate was spun at 10,000 × $g$ for 30 min. The supernatant was incubated for 20 min at 4 °C with 10 mM CaCl$_2$ before loading on a Phenyl Sepharose CL-4B column. The column was washed with a buffer solution containing 10 mM CaCl$_2$ and 50 mM TRIS·HCl (pH 7.5) and then a buffer additionally containing 500 mM NaCl. CaM protein was eluted using a buffer containing 5 mM EGTA, 500 mM NaCl, and 50 mM TRIS·HCl (pH 7.5). SDS-PAGE and Coomassie staining confirmed fractions containing isolated CaM protein. Pooled fractions were dialyzed in 30 mM NaCl and 20 mM MOPS (pH 7.0) and concentrated using an Amicon Stirred Cell with a 3 kDa Ultrafiltration disc. For dye labeling, degassed 0.113 mM CaM protein variant was incubated for 3 h at 22 °C with 0.568 mM AlexaFluor 568 C5 maleimide (ThermoFisher Scientific, cat# A20341) in *labeling* buffer. For binding to a Phenyl Sepharose CL-4B column, 10 mM CaCl$_2$ was added to the labeling reaction before loading on the column. Following washing away the free dye with wash solution (10 mM CaCl$_2$ and 50 mM TRIS·HCl, pH 7.5), labeled protein was eluted by loading a solution containing 5 mM EGTA, 500 mM NaCl, and 50 mM TRIS·HCl (pH 7.5). Fractions containing labeled protein were dialyzed in 30 mM NaCl and 20 mM MOPS (pH 7.0) and stored at −80 °C.

The Ca$^{2+}$-insensitive CaM (CaM$_{1234}$; E-to-A substitutions at 31, 67, 104, and 140) cDNA in pET-7 vector was modified by site directed mutagenesis (QuikChange Lightning kit; Agilent Technologies) to introduce one of four single cysteine mutants (T26C, T34C, Y99C and T110C). The CaM$_{1234}$ variants were expressed, purified, and labeled with AF568 using the method used to purify FKBP12.6[39], as described above. Stoichiometric labeling of fluorescent CaM and FKBP to ≥95%

was demonstrated by the ratio of the absorbance of the bound dye relative to SDS-PAGE densitometry. A lack of unlabeled protein, and the absence of N-terminal initiator methionine in CaM proteins, was confirmed by MALDI-TOF mass spectrometry.

## SR vesicle isolation

SR membrane vesicles were isolated from pig *longissimus dorsi* and pig ventricular tissue by differential centrifugation of homogenized muscle[32]. 'Heavy' SR, rich in RyR, were isolated by fractionation of skeletal 'crude' SR vesicles using a discontinuous sucrose gradient[32]. All vesicles were stored frozen at −80 °C. SR vesicles were stripped of endogenous CaM by incubation with myosin light chain kinase-derived CaM binding peptide, followed by sedimentation in accordance with[53] before radioligand and fluorescence binding assays.

## FKBP12.6 binding studies

The binding of F-FKBP to skeletal SR membranes was measured as previously[39], with exceptions that SR was incubated with 1–60nM F-FKBP, non-specific FRET was measured by excess addition of 5 μM unlabeled WT FKBP12.6, and AF488 labeled FKBP was used. The binding of FKBPs to SR membranes (0.02 mg/ml for skeletal SR and 0.12 mg/mL for cardiac SR) was measured following 90-min incubations in 37 °C buffer containing 150 mM KCl, 20 mM K-PIPES (pH 7.0), 5 mM GSH, 0.1 mg/ml bovine serum albumin, 1 mM EGTA, 65 μM CaCl$_2$ (30 nM free Ca$^{2+}$, calculated using Maxchelator[54]), and 1–60 nM FKBP12.6 labeled at indicated sites. For determinations of nonspecific binding to skeletal SR membranes, 5 μM unlabeled FKBP12.6 was added to the binding buffer. For determination of nonspecific binding to cardiac SR, 20 μM tacrolimus was added to the binding buffer. Bound and free F-FKBP were separated by centrifugation at 100,000 × $g$ for 25 min. Pellets were resuspended in 5% SDS, 50 mM NaCl, 20 mM Na-PIPES (pH 7.0), and 1 mM EGTA. Bound AF488-FKBP12 was determined from the integrated fluorescence intensity from 510 to 610 nm, acquired using a Gemini EM microplate fluorometer (Molecular Devices, Sunnyvale, CA) with excitation at 488 nm and a 495- nm emission long pass filter.

## [³H]ryanodine binding

Skeletal and cardiac SR membranes (1 and 3 mg/ml, respectively) were pre-incubated with the indicated range of [AF568-CaM] for 30 min at 4 °C in a solution containing 150 mM KCl, 5 mM GSH, 1 μg/ml aprotinin/leupeptin, 1 mM DTT, 1 mM EGTA, 65 μM or 1.02 mM CaCl$_2$ (30 nM or 30 μM free Ca$^{2+}$, respectively, as determined by MaxChelator), 0.1 mg/ml of BSA, and 20 mM K-PIPES (pH 7.0). The assay solution at 30 nM Ca$^{2+}$ additionally contained 5 mM Na$_2$ATP and 5 mM caffeine. The binding of [³H]ryanodine (10 or 15 nM; PerkinElmer # NET950025UC) was determined following a 3- h incubation at 37 °C, and filtration through grade GF/B glass microfiber filters (Brandel Inc., Gaithersburg, MD) using a 96-cell Brandel Harvester. In 4 mL of Ecolite scintillation mixture (MP Biomedicals, Solon, OH), the [3H] retained on the filter was measured using a Beckman LS6000 scintillation counter.

## FRET assay preparation

Skeletal or cardiac SR (0.4 mg/ml) membranes were pre-incubated with the donor, 60 nM Alexa Fluor 488-FKBP (D-FKBP), for 90 min at 37 °C in a solution containing 150 mM KCl, 5 mM GSH, 0.1 mg/mL BSA, 1 μg/mL Aprotinin/Leupeptin 1 mM DTT and 20 mM PIPES (pH 7.0). To remove unbound D-FKBP, the SR membranes were spun at 109,760 × $g$. For trilateration and Ca-driven kinetics studies, D-FKBP treated SR membranes (1 mg/mL skeletal SR and 5 mg/mL cardiac SR) were incubated for 60 min at 22 °C with 0.8 μM Alexa Fluor 568-CaM (A-CaM or A-CaM$_{1234}$) in a *FRET buffer* containing 150 mM KCl, 5 mM GSH, 1 μg/mL Aprotinin/Leupeptin, 1 mM EGTA, 2 mM DTT, 65 μM CaCl$_2$ (30 nM free Ca$^{2+}$, as determined by MaxChelator) or 1.02 mM CaCl$_2$ (30 μM free Ca, as determined by MaxChelator) and 0.1 mg/mL of BSA and 20 mM K-PIPES (pH 7.0). For CaM association studies using

a stopped-flow instrument[48,49], one syringe was loaded with loaded with D-FKBP treated SR (2 mg/mL skeletal SR and 6 mg/mL cardiac SR), with the second syringe loaded with 1.6 μM A-CaM, both with *FRET buffer* for either 30 nM or μM free Ca²⁺. For Ca-driven CaM-RyR structural transition studies, one syringe was loaded with D-FKBP treated SR (2 mg/mL skeletal SR and 6 mg/mL cardiac SR) and 1.6 μM A-CaM in *FRET buffer*, with the second syringe loaded with Ca²⁺ or EGTA in *FRET buffer* solution to shift the final free Ca²⁺ from 30 nM to μM Ca²⁺ or 30 μM to nM Ca²⁺ (as determined by MaxChelator).

### Time-correlated single-photon counting for trilateration

For data used in trilateration, we acquired fluorescence decays using time-correlated single-photon counting, as previously[55]. The instrument response function (IRF) was acquired using deionized water in a quartz cuvette (Starna Cells, California, USA). The time-resolved fluorescence decay of samples was measured by time-correlated single photon counting (Becker-Hickl, Berlin, Germany). Following excitation at 485 nm using a sub-nanosecond pulsed diode laser (PicoQuant, Berlin, Germany), filtering the emitted light using a 517/20 nm band-pass filter (Semrock, New York, USA), and detection with a PMH-100 photomultiplier (Becker-Hickl). Global multi-exponential analysis of the fluorescence decay data was used to test a series of structural models, as extensively described previously[55,56]. We determined the number of donor lifetimes and structural states that are present in each sample by fitting a set of models with the number of donor lifetime states, with donor lifetimes increasing from 1 to 4, and the number of structural states increasing from 1 to 4. The final model with three donor lifetimes and three components (donor-only contribution and two distance distributions) was determined by evaluating the dependence of the minimized $\chi^2$ on the number of free parameters in the global model.

### Direct waveform recording for kinetics studies

For kinetics studies, we acquired fluorescence decays using direct waveform recording after rapid mixing with a stopped-flow apparatus, as previously[48,49]. In brief, the instrument uses a Biologic USA SPM/20 single mix stopped-flow accessory coupled to a cuvette in the path of a 473 nm microchip laser with a LD-702 controller (Concepts Research Corporation, Wisconsin, USA) and detection with an 8GS/s digitizer (Acqiris, Switzerland). Emitted light was filtered with a 517/20 nm band-pass filter (Semrock). The dead time for the instrument was 2.0 ms, calibrated using 8-hydroxyquinoline + Mg²⁺ control reaction[57]. After the dead time, the samples were excited every 0.2 ms. For association kinetic studies, 50 waveforms were averaged for every 10 ms. For Ca²⁺-driven structural kinetic studies, 5 waveforms were averaged for every ms. The instrument response function (IRF) was acquired using deionized water in the cuvette. Fluorescence waveforms were fitted based on a one-exponential decay function using least-squares minimization global analysis software (Fluorescence Innovations, Inc.). The FRET efficiency (*E*) was determined as the fractional decrease of donor fluorescence lifetime ($\tau_D$), due to the presence of acceptor fluorophore ($\tau_{DA}$), using the following equation:

$$E = 1 - \frac{\tau_{DA}}{\tau_D} \qquad (1)$$

### Simulation of donor probe positions

To interpret the FKBP-CaM FRET data, we determined the location of the donor AF488 probes when attached to FKBP in RyR1 and RyR2. This was done similarly to what we had previously done[42] except that here we used the higher-resolution RyR structures currently available. For the computational studies of RyR1, the structure chosen was the 3.6-Å closed pore state (PDB ID: 5T15) and the corresponding cryo-EM density map (EMDB ID: 8342)[1]. This structure was chosen due to its

high resolution and experimental conditions that are similar to our 30 μM Ca²⁺ condition. The high resolution is important to resolve density in the DR2 region near FKBP. For RyR2, we used the 3.6-Å structure corresponding to the closed state (PDB ID: 6IJ8) and the corresponding cryo-EM density map (EMDB ID: 9833)[4].

For molecular simulations of the donor probes, we need to take into account the protein volume corresponding to the region around FKBP. Near FKBP is the DR2 region, which, due to disorder, has not been resolved in any cryo-EM structures for RyR. To allow for that region, we used the protein density of the cryo-EM maps represented as dummy atoms in the simulations. The grid points of the new map were converted to dummy atoms using the Situs software package "vol2pdb" program[58] and saved in a pdb file. Grid-points overlapping with the coordinates of FKBP were removed. To speed up calculations, only dummy atoms within 30 Å from FKBP were used to restrict the conformational space of the fluorescent probe, and all grid-points that are completely buried were deleted.

In the atomic structure of FKBP12.6, the eight residues at positions 1, 6, 14, 32, 44, 49, 65, and 85 that were used for probe attachment in the fluorescence experiments were mutated to cysteines using Discovery Studio Visualizer 2019 (BIOVIA, Dassault Systèmes, San Diego, CA). AF488 was attached, respectively, to each of the eight cysteines, generating eight starting conformations for simulated annealing using the software package CNS 1.3[59]. The simulated annealing protocol has been described in detail and validated previously[42]. With the assumption that the fluorescence probe undergoes fast dynamics in relation to the fluorescence lifetime[60], means that we can use the average position of the chromophore's center, as calculated from simulated annealing.

### Calculation of predicted acceptor probe locations

To calculate the expected distances between donor and acceptor probes in the cryo-EM structures we developed a new fast method to determine the predicted location of the acceptor probes on CaM. As this information is just used for comparison with experimental FRET distance it would not be necessary to run the computationally costly simulated annealing calculations to determine the probe locations. Data from the previous probe simulations using simulated annealing of AF488 attached to FKBP were analyzed for the distance between the center of the chromophore back to the Cβ atom of the Cys which the probe is attached. These distances were binned to generate a probability distribution. The probe center to the Cβ atom distance was found to be between 8 and 20 Å with an average value of ~15 Å. A region in space that the probe can sample was determined by selecting points that did not clash with atoms of the protein around the attachment site of the probe, from points on concentric spheres of evenly spaced points. A radius of 6 Å for the probe was used. These selected points in space were combined with the probability distribution previously determined to calculate the average probe location. The method was validated by calculating average probe positions for AF488 on FKBP and was found to be quite accurate as it could predict probe locations that were on average within 2-3 Å from the locations determined by simulated annealing.

### Trilateration

From the simulated-annealing calculations above, we obtained coordinates for the probe locations, which we used as effective AF488 chromophore locations for trilateration. We used these coordinates for each of the eight FKBP attachment sites (1, 6, 14, 32, 44, 49, 65, and 85), in combination with distances calculated from FRET, to determine a locus in space corresponding to the acceptor attached to each of the four labeled sites on CaM (26, 34, 99, and 110). We have developed software that performs trilateration using an arbitrary number of donor sites and distances. Our new trilateration approach takes into account Gaussian distance distributions as fitted to the FLT-FRET measurement for each donor position. The normalized sum of the

distance distributions from all the donor sites to one acceptor is generated as a probability map describing the acceptor's location. This volumetric map is saved in MRC/CCP4-format, which can be displayed using common molecular graphics software. To focus the results to just the region near RyR the probability map was masked to the region within 20 Å from the RyR surface using UCSF Chimera 1.16[61]. Chimera was also used to calculate the volume and center positions of the trilaterated loci and determine the iso value setting for displaying the loci at the selected 1000 Å$^3$ volume. Molecular structure figures were created using VMD 1.9.3[62].

## Statistics and reproducibility

Sample means are typically from three or more independent experiments; numbers of observations (n) per independent experiment undertaken on separate days are given in the figure legends or the figure. No statistical method was used to predetermine the sample size. Each experiment was repeated using at least two independent SR membrane preparations, isolated from different animals. The experiments were not randomized. The Investigators were not blinded to allocation during experiments and outcome assessment. Analysis was only undertaken on data from experiments when expected result from control conditions was obtained. Average data are provided as mean ± SD. Statistical significance was evaluated by use of unpaired Student's t-test as indicated for each figure.

## Reporting summary

Further information on research design is available in the Nature Portfolio Reporting Summary linked to this article.

## Data availability

Cryo-EM structure data used in this study were obtained from the PDB and EMDB databanks. The following were used: 6JI8, EMD-9833, 6JV2, EMD-9889, 5T15, EMD-8342, 6X32, 7TZC. Data generated or analyzed during this study are included in this article (and its supplementary information files) as completely as practically possible. Due to the large file size, all data, including simulation trajectories, probability maps, and fluorescence lifetime FRET data, are available from the corresponding author on request, all requests will be answered within 4 weeks. Source data are provided with this paper.

## Code availability

Code to process CaM-RyR trilateration is available at: https://sourceforge.net/projects/trilat-umn/.

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

## Acknowledgements

This work was supported by American Heart Association Postdoctoral Fellowship [16POST31010019] (RTR), NIH grants R01HL092097 (R.L.C., and D.M.B.), R01AR082533 (D.D.T., R.L.C., and R.T.R.), R01AR068431 (M.S.) and R01HL139065 (D.D.T., R.L.C., and R.T.R) Spectroscopy studies reported in this article were performed at the Biophysical Technology Center, University of Minnesota Department of Biochemistry, Molecular Biology, and Biophysics.

## Author contributions

R.T.R. designed research, performed research, analyzed data, and drafted manuscripts. B.S. wrote code, analyzed data, generated structural models, and drafted manuscripts. J.Y.Z. performed research. M.S. provided comparison data and edited the manuscript. D.D.T. edited the manuscript. D.M.B. designed the research and edited the manuscript. R.L.C. designed the research and edited the manuscript.

## Competing interests

All authors declare no competing interests. R.L.C. is currently an employee of the National Institutes of Health. This work was conducted during his previous employment, at the University of Minnesota – Twin Cities. The opinions expressed in this article are the author's own and do not reflect the view of the National Institutes of Health, the Department of Health and Human Services, or the United States government.
