## [Peer Review File · Nature Communications]

REVIEWER COMMENTS

Reviewer #1 (Remarks to the Author):

In the manuscript entitled “Kinetics and Mapping of Ca-driven Calmodulin conformations on Skeletal and Cardiac Muscle Ryanodine Receptors”, Rebbeck et al. provide unique insight into the Ca-driven CaM conformations on both RyR1 and RyR2 and CaM-RyRs binding kinetics, under physiologically relevant Ca condition. However, I have several significant concerns that should be addressed before publication.

Critique

1. It is crucial the mature CaM without the initial Met is used, any artefacts at the N-terminus have the potential to change the regulatory effects of CaM on RyRs. From the METHODS, it is unclear that how to purify the CaM samples. Please add the details for the purification of CaM and clarify that the WT-CaM is the mature CaM.

2. Line 140, the authors state that “With the binding of FKBP12.6 very similar on RyR1 and RyR2, it is reasonable to assume that, like RyR1, labeling at any of these eight sites does not impact FKBP12.6-RyR2 binding”, which is an inaccurate expression. Actually, it has been reported that FKBP12.6 can increase RyR1 activity but inhibit RyR2 activity, although they are located in a similar site on RyR1 and RyR2. In addition, FKBP12.6 and FKBP12 share 85% sequence homology and have similar 3D structures, but they show markedly different affinities for binding to RyR2.

Specific comments :

1. Line 68, RyR-apo-CaM should be RyR1-apo-CaM.
2. Line 66, The upper case or lower case for the “Helical and Handle domain” should be consistent throughout the manuscript.
3. Figure 2: As low Ca²⁺ may induce confusion, I recommend revise the “RyR2 at low Ca²⁺” as “under EDTA condition”.
4. Line 424 : For clarity, please add details of purification of FKBP12.6 and avoid using “... as described previously” without detailed information throughout the METHODS.

Reviewer #2 (Remarks to the Author):

In this manuscript Rebbeck et al. analyzed dynamics of calmodulin conformations interacting with ryanodine receptors by lifetime measurement of FRET donor. By introducing Cys residue to various positions of FKBP and CaM, they labeled those proteins with AF488 and AF568 as a FRET donor and acceptor, respectively. The authors have previously reported the idea of detecting the conformational change of CaM interacting with RyR by FRET, but in this paper, they extended the labeling sites on both FKBP and CaM. More importantly, they analyzed the conformational change in 0.2-ms resolution by measuring fluorescence lifetime of the donor by employing the direct waveform detection technique. It highlighted the differences in dynamic interaction of CaM to RyR1 and RyR2, as well as the potentially different structural states from cryo-EM models. This reviewer found the approach unique and highly potential for further investigations of the RyR modulators. For these reasons the reviewer think that the manuscript is worth publishing in Nature Communications. The authors should address the following points prior to the publications.

1) The authors are discussing rate of CaM binding with the observed time constant. In theory, the observed association time constant is reciprocal of the observed association rate constant. The observed rate constant is the sum of the intrinsic dissociation rate constant and the product of intrinsic association rate constant and ligand concentration (refer the following equation).

$$1/\tau_{\text{obs}} = k_{\text{on}} \times [\text{CaM}_{\text{free}}] + k_{\text{off}}$$

Therefore, the CaM concentration and the off-rate constant also influenced the observed time constant. The interpretation of the time constant should be described more carefully in the text.

2) Details of the direct waveform detection is missing although this is a key of the measurement. It has been published previously, but including minimum information for the measurement is appropriate.

3) TCSPC was also used for the mapping of the CaM position (Fig. 3). Details of the measurements should be involved.

Reviewer #3 (Remarks to the Author):

The manuscript by Rebbeck et al. reported mapping of Ca²⁺ driven calmodulin conformations on RyR1 and RyR2 by fluorescence lifetime detection of FRET (FLT-FRET). The positions and orientations of apo-CaM revealed by FLT-FRET agrees with the results of published cryo-EM structures, but more potential structural states of RyR under Ca²⁺ CaM bound condition was discovered in this study. Compared to their previous work, the authors improves the spatiotemporal resolution of the FLT-FRET trilateration method, which better reveals the dynamic conformational changes induced by Ca²⁺ and CaM binding in two isoforms of RyRs that could not be easily captured by cryo-EM. The manuscript is clear and well written. I support the publication of the paper provided that the minor comments below are addressed.

1. The long second distance is explained as the FRET signal from A-CaM of the neighboring subunit. In principle, there are two neighboring subunits, both of which could give different FRET signals. Why was the “third distance” not observed?
2. When the second distance is comparable to R₀, can one get the ratio between different conformations based on the distribution? The authors can cross validate these quantified distribution results by comparing them with the 3D-classification results of the related cryo-EM structures, and add this part in result and discussion.
3. The authors can use their distance as constraints to run a molecular dynamics simulation and model the potential new structural states of RyR under Ca²⁺ CaM bound condition. Add a supplementary figure to show the structures of the new states.
4. A figure to illustrate the time-resolved structural change would be useful for the readers to understand the overall kinetics of the conformational change. The current figure 7 misses the information of time-resolution. Alternatively, for a better illustration, movies generated from MD as suggested in comment 3 showing the dynamic structural changes and the kinetics information would also be useful.

Dear Review panel,

We appreciate all of the thoughtful and constructive comments on our manuscript (NCOMS-23-43797). We have carried out further experimental work and revised the text and figures to address each of your concerns and suggestions. This has led to a much improved manuscript, which we now submit for your consideration. Our point-by-point response to each comment is presented below. Significant changes to the text are indicated in **blue font** in the revised manuscript, and appropriately indicated in the responses below.

Peer Reviewer #1:

Comment summary: *In the manuscript entitled “Kinetics and Mapping of Ca-driven Calmodulin conformations on Skeletal and Cardiac Muscle Ryanodine Receptors”, Rebbeck et al. provide unique insight into the Ca-driven CaM conformations on both RyR1 and RyR2 and CaM-RyRs binding kinetics, under physiologically relevant Ca condition. However, I have several significant concerns that should be addressed before publication.*

Comment 1: *It is crucial the mature CaM without the initial Met is used, any artefacts at the N-terminus have the potential to change the regulatory effects of CaM on RyRs. From the METHODS, it is unclear that how to purify the CaM samples. Please add the details for the purification of CaM and clarify that the WT-CaM is the mature CaM.*

Response: To directly address this, we have revised our *METHODS* section *Expression, purification and fluorescent labeling of FKBP and CaM* text to explain our procedure for purifying CaM samples. It is our standard procedure to use mass spectroscopy to confirm that the initial Met has been cleaved during expression in *E. coli*. This is done with each batch prepared.

Comment 2: *Line 140, the authors state that “With the binding of FKBP12.6 very similar on RyR1 and RyR2, it is reasonable to assume that, like RyR1, labeling at any of these eight sites does not impact FKBP12.6-RyR2 binding”, which is an inaccurate expression. Actually, it has been reported that FKBP12.6 can increase RyR1 activity but inhibit RyR2 activity, although they are located in a similar site on RyR1 and RyR2. In addition, FKBP12.6 and FKBP12 share 85% sequence homology and have similar 3D structures, but they show markedly different affinities for binding to RyR2.*

Response: This is a fair point, and we do acknowledge the potential complexity of differential functional regulation of the RyRs by FKBP. To directly address the reviewer’s comment, we have conducted extensive additional control studies to measure the D-FKBP binding profiles for **both** RyR1 and RyR2 by carrying out co-sedimentation saturation binding experiments for the entire set of D-FKBP used in this project. These results are summarized in the updated Supplementary Fig. 1D and text in *RESULTS* section *Labeling FKBP12.6 and CaM does not alter binding to RyR* paragraph 2. Additionally, the new procedure was incorporated into the *METHODS* section *FKBP12.6 binding studies*. Overall, the affinities of the different AF488 labeled FKBP12.6 variants were similar between RyR1 and RyR2. Based on our new data, we can report that AF488-labeling of FKBP12.6 at any of the eight sites does not impact FKBP12.6-RyR2 binding.

Specific comment 1: *Line 68, RyR-apo-CaM should be RyR1-apo-CaM.*

Response: Text revised. Thank you for pointing this out.

Specific comment 2: *Line 66, The upper case or lower case for the “Helical and Handle domain” should be consistent throughout the manuscript.*

Response: Text revised. Thank you for pointing this out.

Specific comment 3: *Figure 2: As low Ca²⁺ may induce confusion, I recommend revise the “RyR2 at low Ca²⁺” as “under EDTA condition”.*

Response: Thank you for pointing out potential confusion. Low Ca is not clear. We adjusted the text in Figure 1 legend. For our results, we prefer not to change the text to “under EGTA conditions”, as all Ca²⁺ concentrations are buffered by EGTA. Instead, we changed the text to “at nM Ca²⁺” or for the higher Ca²⁺ level “at μM Ca²⁺”.

Specific comment 4: *Line 424 : For clarity, please add details of purification of FKBP12.6 and avoid using “... as described previously” without detailed information throughout the METHODS.*

Response: Further details were added to the *METHODS* section *Expression, purification and fluorescent labeling of FKBP and CaM* paragraph 1. Thank you for pointing this out.

Peer Reviewer #2:

Comment summary: *In this manuscript Rebbeck et al. analyzed dynamics of calmodulin conformations interacting with ryanodine receptors by lifetime measurement of FRET donor. By introducing Cys residue to various positions of FKBP and CaM, they labeled those proteins with AF488 and AF568 as a FRET donor and acceptor, respectively. The authors have previously reported the idea of detecting the conformational change of CaM interacting with RyR by FRET, but in this paper, they extended the labeling sites on both FKBP and CaM. More importantly, they analyzed the conformational change in 0.2-ms resolution by measuring fluorescence lifetime of the donor by employing the direct waveform detection technique. It highlighted the differences in dynamic interaction of CaM to RyR1 and RyR2, as well as the potentially different structural states from cryo-EM models. This reviewer found the approach unique and highly potential for further investigations of the RyR modulators. For these reasons the reviewer think that the manuscript is worth publishing in Nature Communications. The authors should address the following points prior to the publications.*

Comment 1: *The authors are discussing rate of CaM binding with the observed time constant. In theory, the observed association time constant is reciprocal of the observed association rate constant. The observed rate constant is the sum of the intrinsic dissociation rate constant and the product of intrinsic association rate constant and ligand concentration (refer the following equation).*

$$1/\tau_{obs} = k_{on} \times [CaM_{free}] + k_{off}$$

Therefore, the CaM concentration and the off-rate constant also influenced the observed time constant. The interpretation of the time constant should be described more carefully in the text.

Response: Indeed, the reviewer is correct. We have revised the text in *RESULTS* section *Kinetics of CaM-RyR binding using stopped-flow FLT-FRET* (paragraph 2) to acknowledge that we use our measured time constant values as indicative of association kinetics as follows: “Given that the CaM binding to RyR1/2 measured in Fig. 5 is very much slower than the dissociation (by >30-fold)^{28, 51} these time constants for CaM binding (at 800 nM CaM) are closely indicative of CaM-RyR association kinetics (e.g. $k_{on} \sim 75 \times 10^6 \text{ M}^{-1}\text{min}^{-1}$ for RyR2).”

Rearranging and using k_{off} at 50 nM [Ca] from Yang et al. 2014 (0.1 min^{-1}) and $k_{obs} = 1 \text{ s}^{-1} = 60 \text{ min}^{-1}$):

$$k_{on} = (k_{obs} - k_{off})/[CaM] = (60 \text{ min}^{-1} + 0.1 \text{ min}^{-1})/0.8 \mu\text{M} = 75.13 \times 10^6 \text{ M}^{-1} \text{ min}^{-1}$$

Thus, the k_{obs} dominates the numerator (600-fold). That dominance will be much higher at $30 \mu\text{M Ca}^{2+}$, where k_{off} is very much slower.

Comment 2: *Details of the direct waveform detection is missing although this is a key of the measurement. It has been published previously, but including minimum information for the measurement is appropriate.*

Response: That is a fair point. We have greatly expanded the *METHODS* section *Direct waveform recording for kinetics studies* to be more complete

Comment 3: TCSPC was also used for the mapping of the CaM position (Fig. 3). Details of the measurements should be involved.

Response: Thanks. Again, we expanded the text in the *METHODS* section *Time-correlated single-photon counting for Trilateration* to make this clearer.

Peer Reviewer #3:

Comment summary: *The manuscript by Rebbeck et al. reported mapping of Ca²⁺ driven calmodulin conformations on RyR1 and RyR2 by fluorescence lifetime detection of FRET (FLT-FRET). The positions and orientations of apo-CaM revealed by FLT-FRET agrees with the results of published cryo-EM structures, but more potential structural states of RyR under Ca²⁺ CaM bound condition was discovered in this study. Compared to their previous work, the authors improves the spatiotemporal resolution of the FLT-FRET trilateration method, which better reveals the dynamic conformational changes induced by Ca²⁺ and CaM binding in two isoforms of RyRs that could not be easily captured by cryo-EM. The manuscript is clear and well written. I support the publication of the paper provided that the minor comments below are addressed.*

Comment 1: *The long second distance is explained as the FRET signal from A-CaM of the neighboring subunit. In principle, there are two neighboring subunits, both of which could give different FRET signals. Why was the “third distance” not observed?*

Response: Based on the cryo-EM derived models, a third distance would be beyond the detection of FRET (~20-120 Å) both for the N- and the C-lobes of CaM. We have revised the text to explain how the second distance, when present, is a result of A-CaM existing in a range of partially populated states. In this manuscript we have focused on locating the most populated state. We have better clarified this point in the *RESULTS* section *Resolving distance distribution between probes from FLT-FRET experiments* paragraph 2.

Comment 2: *When the second distance is comparable to R₀, can one get the ratio between different conformations based on the distribution? The authors can cross validate these quantified distribution results by comparing them with the 3D-classification results of the related cryo-EM structures, and add this part in result and discussion.*

Response: Yes, if both distances are comparable to the R₀, one can get the ratio between different conformations based on the distribution. Unfortunately, we are unable to cross validate these with the current cryo-EM results. So far, the published cryo-EM structures of RyR with bound CaM have revealed CaM either in the Ca-CaM or the apo-CaM position, depending on the Ca²⁺ concentration. In this context, and as the reviewer indicates, this is where our studies provide a unique contribution, hinting at intermediate binding states. Additionally, there are no published cryo-EM structures of RyR with CaM bound solved in any specific state that show different classes of particles with different CaM binding. Further work by cryo-EM researchers may reveal additional structures and possible alternative or intermediate CaM binding sites. When such information becomes available, the suggested type of analysis of the FRET data may become feasible.

Comment 3: *The authors can use their distance as constraints to run a molecular dynamics simulation and model the potential new structural states of RyR under Ca²⁺ CaM bound condition. Add a supplementary figure to show the structures of the new states.*

Response: Molecular dynamics (MD) simulations might provide plausible molecular models constrained by the FRET data. However, running MD simulation and protein-protein docking on the extremely large RyRs is a considerable endeavor, which must be conducted as an independent project. The starting point could be the cryo-EM structures of RyR with bound CaM. In addition, these structures have large regions near the CaM binding region that are unresolved. Especially, the loops connecting the helices in the Helical

domain 1 are unresolved. Helical domain 2 is largely unresolved, probably due to significant disorder. With the current information, simply modeling the unresolved regions of RyR and running an MD simulation would be a resource-intensive undertaking, and provide equivocal outcomes at best. Considering this, we believe that this is unfeasible at this stage. We do present the possible locations of the individual lobes of CaM bound to RyR in nM and μM Ca^{2+} here, but our data is available (in tables of the SI section of this paper) so computational groups could use that for guided MD simulations to gain insight on potential intermediate loci for CaM binding to RyR. A statement about such possible future developments has been added to the last paragraph in the *DISCUSSION* section.

Comment 4: *A figure to illustrate the time-resolved structural change would be useful for the readers to understand the overall kinetics of the conformational change. The current figure 7 misses the information of time-resolution. Alternatively, for a better illustration, movies generated from MD as suggested in comment 3 showing the dynamic structural changes and the kinetics information would also be useful.*

Response: We added two cartoons to Fig. 6 (new panels C and F) that illustrate the differences in Ca-driven kinetics between the conditions tested. As to generating a movie from MD, please see our response to Comment 3.

REVIEWERS' COMMENTS

Reviewer #1 (Remarks to the Author):

I am satisfied by the corrections to the manuscript and recommend publication.

Reviewer #2 (Remarks to the Author):

The authors properly addressed to the comments from this reviewer. The reviewer recommend the manuscript to be published.

Reviewer #3 (Remarks to the Author):

All my concerns have been addressed in the revision. I recommend its publication.